# A Review of Life Prediction Methods for PEMFCs in Electric Vehicles

**Aihua Tang** [1] , **Yuanhang Yang** [1] **, Quanqing Yu** [2],* , **Zhigang Zhang** [1] **and Lin Yang** [1]

[1]  School of Vehicle Engineering, Chongqing University of Technology, Chongqing 400054, China
[2]  School of Automotive Engineering, Harbin Institute of Technology, Weihai 264209, China
*   Correspondence: qqyu@hit.edu.cn

**Abstract:** The proton-exchange membrane fuel cell (PEMFC) has the advantage of high energy conversion efficiency, environmental friendliness, and zero carbon emissions. Therefore, as an attractive alternative energy, it is widely used in vehicles. Due to its high nonlinearity, strong time variation, and complex failure mechanisms, it is extremely difficult to predict PEMFC life in electric vehicles. The uncertainty of life predictions for the PEMFC limits its wide application. Since it is particularly important to accurately carry out PEMFC life predictions, significant research efforts are directed toward tackling this issue by adopting effective methods. In this paper, a number of PEMFC life prediction methods for electric vehicles are reviewed and summarized. The goal of this review is to render feasible and potential solutions for dealing with PEMFC life issues considering dynamic vehicle conditions. Based on this review, the reader can also easily understand the research status of PEMFC life prediction methods and this review lays a theoretical foundation for future research.

**Keywords:** failure mechanisms; PEMFC; life prediction methods; dynamic vehicle conditions





## 1. Introduction

In recent years, new energy vehicles have attracted increased attention worldwide because of the advocacy of green, environmentally friendly, and low-carbon transportation modes. Consistent with development trends in renewable energy, hydrogen-based technologies, such as the PEMFC, which is widely used as fuel in electric vehicles, are increasingly regarded as key to promoting transport electrification [1]. Compared with traditional internal combustion engine vehicles, the PEMFC has higher efficiency and zero emissions [2]. With the popularization of the PEMFC in electric vehicles becoming a trend, more and more scholars are paying attention to its related research. During the operation of the PEMFC, its performance degradation can be caused by many factors such as external complex working conditions, design processes, and materials. Early performance degradation is mainly due to the loss and aggregation of catalysts, whereas later performance degradation is mainly caused by structural damage to the membrane's electrode assembly [3]. Lifetime uncertainty caused by performance degradation makes life predictions for the PEMFC challenging. However, inaccurate PEMFC life predictions could lead to the failure of fuel cells in electric vehicles and even accidents during driving. Many researchers have summarized the development process, advantages, disadvantages, and prospects of PEMFC life prediction methods [4–6]. Therefore, many scholars have developed innovative life prediction algorithms for comparative experiments to improve prediction accuracy and reliability [7]. Some researchers have analyzed the test durability conditions, methods, and health indicators of the PEMFC compared to test data under different working conditions [8] and have highlighted the importance of life prediction methods, ascertaining that life prediction is the key to determining whether or not the PEMFC can be widely used in electric vehicles. Failure mechanisms have a certain influence on the accuracy of life prediction. Factors such as temperature changes and improper battery clamping processes could cause physical degradation. Factors such as the production of peroxide

in the reaction process could cause chemical degradation of the battery. The complexity of the failure mechanism could cause great problems for life predictions. At the same time, the degradation index of the proton-exchange membrane fuel cell also determines the accuracy and speed of life predictions, which can be divided into degradation indexes based on measured data such as stack voltage, polarization curve, ohmic resistance, and the degradation indexes of components that can reflect the internal degradation trend, so as to help users understand the aging state of the PEMFC and take timely maintenance measures to prolong its service life [9]. In addition, some scholars have established a degradation model with the derived thickness of the polarization curve as the index. Based on this model, the developed life prediction and estimation algorithm achieves high accuracy and strong robustness [10]. Some scholars have proposed a fusion prediction strategy, which uses a degradation model to deal with the dynamic operating conditions of the PEMFC and have extracted the degradation index for the prediction [11]. The establishment and application of degradation indicators could also promote the development of more life prediction methods, which are broadly classified into three categories [12,13]: data-driven approaches, model-driven approaches, and hybrid approaches, as shown in Figure 1. A large number of studies on life prediction methods also show the development trends and challenges of the PEMFC in this area [14–16]. The reliability of long-term life predictions under complex conditions [17] and the influence of different factors on dynamic operating condition predictions [18,19] need to be further explored.

In Section 2, the data-driven approaches are described. In Section 3, the model-driven approaches are summarized, which are divided into three types, the filter model, degradation mechanism model, and empirical model. In Section 4, the hybrid approaches are discussed. In Section 5, the future trends and challenges are discussed. In this paper, the life prediction methods of the PEMFC in recent years are classified and summarized so that readers have a clearer understanding of the life prediction methods, and the future development directions of the life prediction methods are summarized, which has guiding significance for subsequent experimental research.

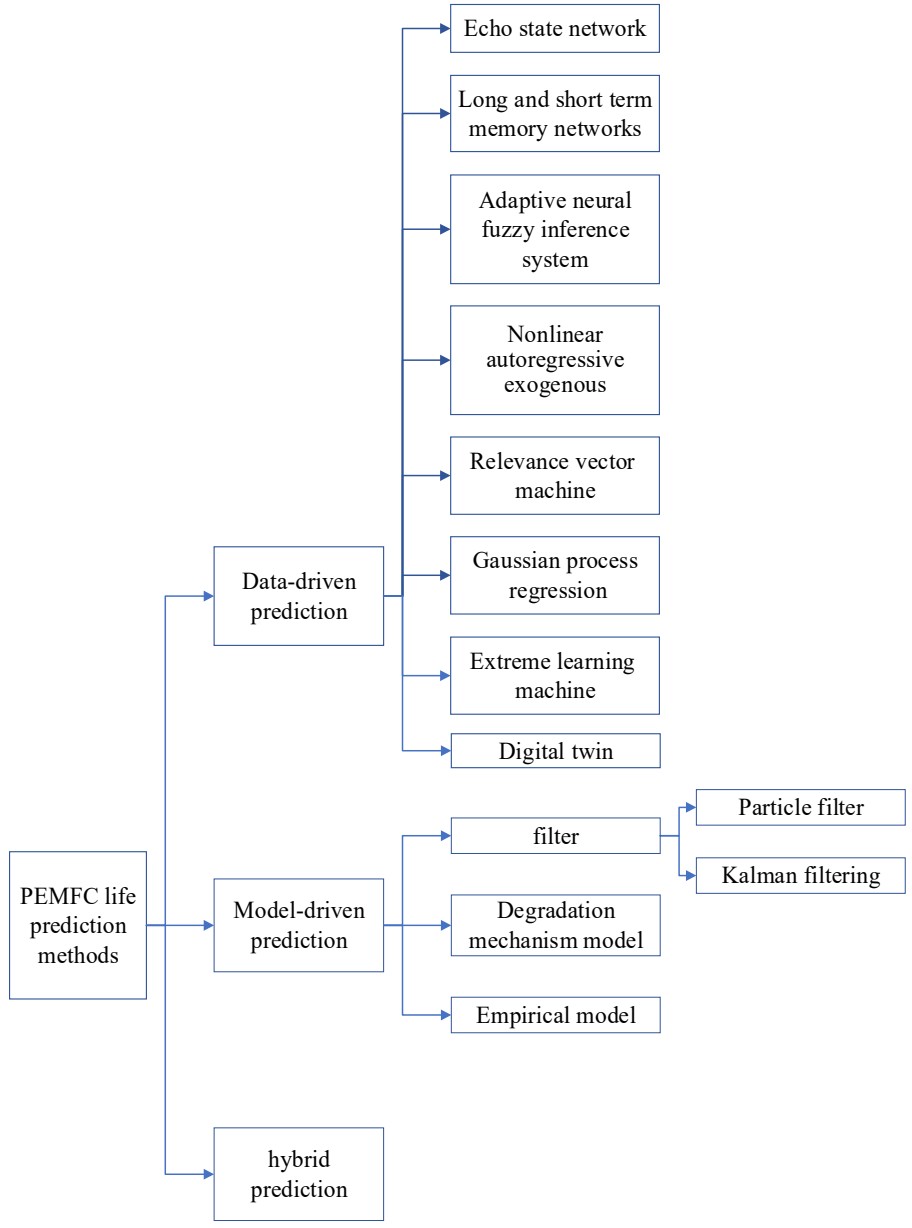

**Figure 1.** An overview of PEMFC life prediction methods.

## 2. Data-Driven Approaches

Due to the nonlinear, strong time-varying, and high coupling characteristics of the PEMFC, difficulties in life prediction are aggravated. To tackle this challenge, data-driven methods are usually used to learn and intelligently provide valuable information from the current online sampling data and the large amount of historical offline data stored in the system [20], This avoids overdependence on the complex decay mechanism of the PEMFC [21]. The data-driven approaches can be utilized to monitor the health status of the PEMFC system by learning and training data [22–26]. Data-driven approaches include echo state network (ESN), long- and short-term memory network (LSTM), adaptive neuro-fuzzy inference system (ANFIS), nonlinear autoregressive exogenous (NARX), relevance vector machine (RVM), Gaussian process regression (GPR), extreme learning machine (ELM), and Digital twin.

### 2.1. Echo State Network

As a neural network with wide application fields, the ESN is used for the system parameter identification of known dynamic systems [27]. Compared with the classical neural network, the ESN is simpler and more efficient and can be locally optimized. The ESN training data can be carried out in two steps: sampling and weight calculation. The ESN trains the weights from the hidden layer to the output layer, transforms the nonlinear problem into a linear regression problem, simplifies the calculation, and accelerates the convergence speed. It can perform multi-step predictive tasks and model predictive control [28].

Rania Mezzi [29] developed a novel method based on the ESN to forecast battery life, which can be predicted in the case of an unknown variable load profile. The experimental results showed that the proposed algorithm obtains high PEMFC residual life prediction accuracy under uncertainty. Meiling Yue [30] proposed an adaptive data-driven PEMFC prediction method based on multiplicative feature decomposition and the ESN, which was used to extract the aging tendency from the experimental data. Furthermore, it could be adapted to train the behavior model and predict the future state. In addition, this method could be employed to forecast the degradable behavior of the PEMFC under dynamic operating conditions. Zhiguang Hua [31] introduced the moving weight matrix method and verified the feasibility and effectiveness of the ESN under different configurations for life predictions in different periods. The output weight array could be updated in real-time with the continuous movement of the training date. It was found that the different structures of the ESN will affect the prediction accuracy and prediction time of PEMFC life predictions. Zhiguang Hua et al. [32] proposed to predict residual life using multiple-input multiple-output ESN (MIMO-ESN) combined with multiple aging indicators such as heap voltage and heap current. A multi-input ESN was designed and tested for more than 1000 h. Finally, the proposed MIMO-ESN method achieved a high prediction accuracy of residual life and generalization ability by experimental verification.

### 2.2. Long- and Short-Term Memory Network

The LSTM is a special recursive neural network that can solve time-series prediction problems through the recurrent neural network. An LSTM for the PEMFC was proposed by Jiawei Liu [33] using uniformly spaced sampling and locally weighted regression discrete smoothing technology to rebuild and smooth the data, which required lower costs while accurately predicting the remaining life. A long-term aging experiment of the PEMFC showed that this method had higher prediction accuracy than a backpropagation neural network (BP) and was suitable for online predictions but can only predict the working conditions of the constant load operation. The prediction block diagram based on the LSTM is shown in Figure 2 [33].

A navigation sequence-driven LSTM (NSD-LSTM) was designed by Zhu Wang et al. [34] to solve the cumulative error and uncertainty of the LSTM model due to long-term identification. Then, the method was verified using different data sets. Compared with the NARX and ESN data-driven methods, the NSD-LSTM has a more accurate life prediction ability.

### 2.3. Adaptive Neural Fuzzy Inference System

ANFIS is a fuzzy reasoning system implemented under the framework of an adaptive network [35]. In an ANFIS system, the nonlinear function is utilized to model, nonlinear components are identified online, and time series are predicted. ANFIS combines a neural network with a fuzzy system to increase the influence of logic and prior knowledge and employs an artificial neural network to learn the membership function of fuzzy logic [36]. S. Rezazadeh [37] used ANFIS to simulate the performance of the PEMFC and compared it with the experimental results of the training and test data. Moreover, the performance of the PEMFC could be modeled utilizing these data with ANFIS. The results showed that the proposed ANFIS modeling method was feasible and could be used for PEMFC life predictions. The ANFIS structure diagram can be seen in Figure 3 [37].

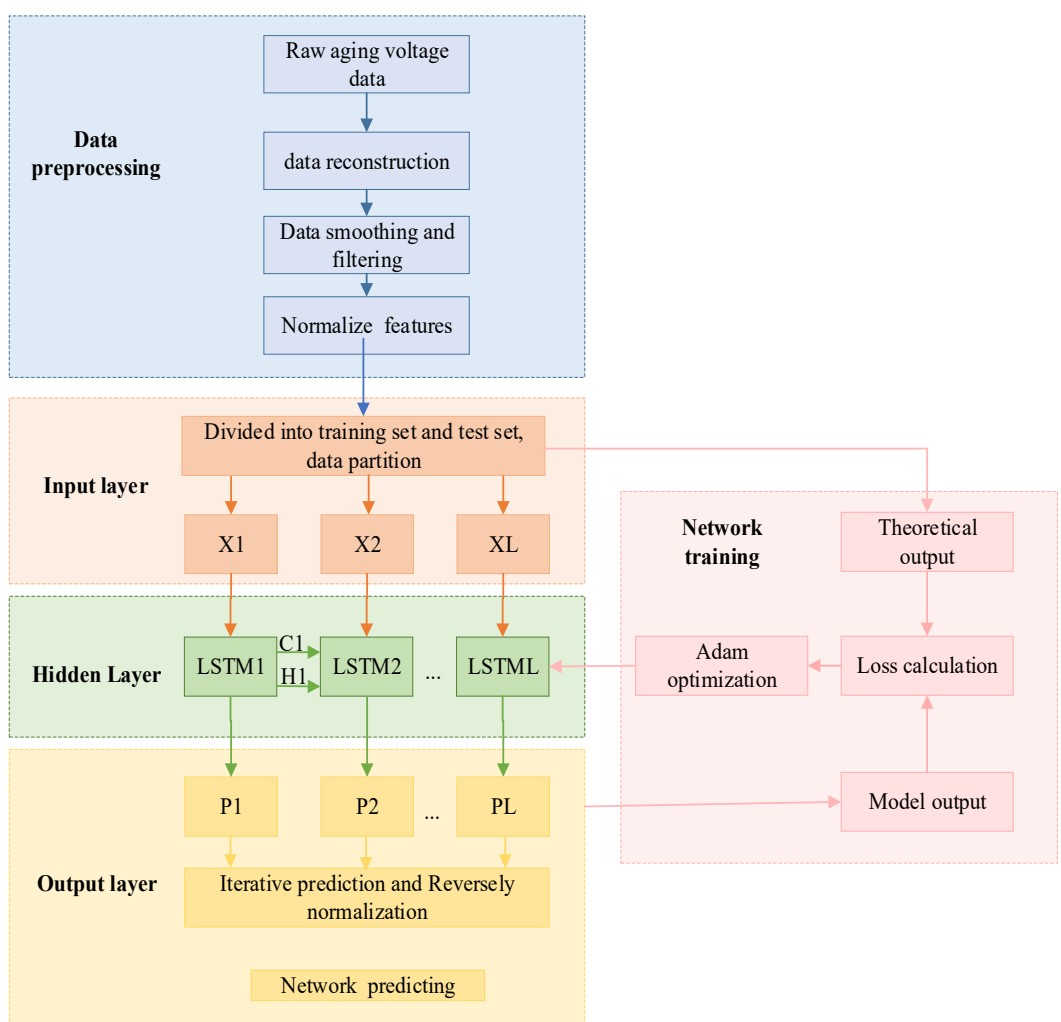

**Figure 2.** PEMFC aging forecasting algorithms based on LSTM network.

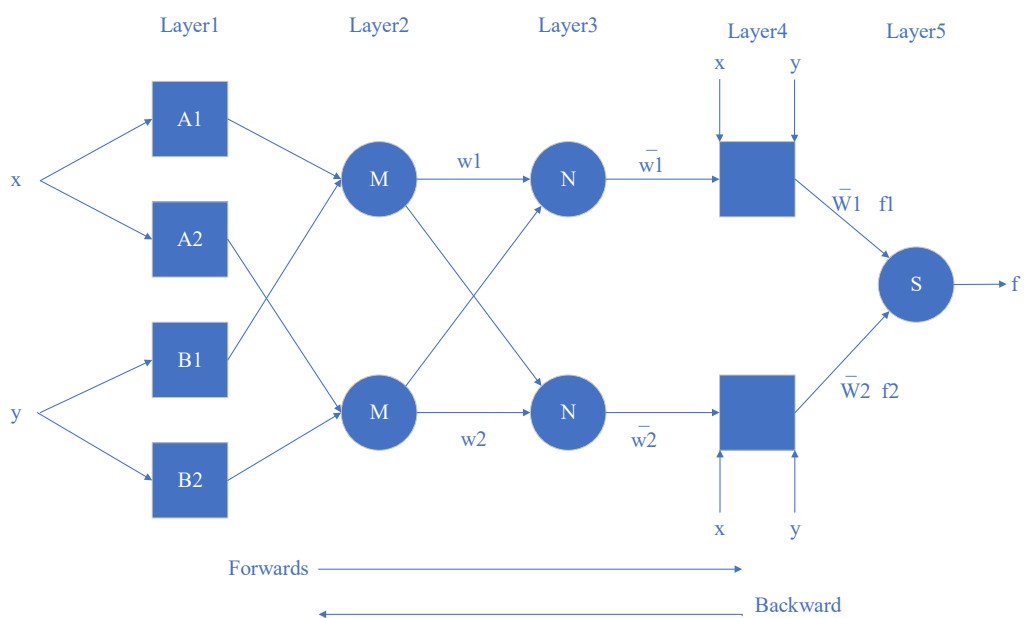

**Figure 3.** Architecture structure diagram of ANFIS.

### 2.4. Nonlinear Autoregressive Exogenous Neural Network

Kui Chen et al. [38] fused wavelet analysis and NARX to predict the life of the PEMFC. The reliability of the model was verified by three durability experiments of the PEMFC under different conditions and the accuracy of the degradation prediction was higher than that of the traditional NARX. Francisco da Costa Lopes [39] proposed a system identification modeling method based on NARX and the nonlinear output error neural nets to achieve the PEMFC stack model for accurate prognostication. The constructed model is utilized to provide high-precision stack voltage predictions for a long time. In addition, the experimental results proved that the model developed using NARX and nonlinear output error (NOE) neural structures can well approximate the time-varying behavior of the PEMFC battery pack without retraining the network for a long time.

### 2.5. Relevance Vector Machine

The RVM constructs a learning machine based on the Bayesian framework, which has a shorter prediction time and better generalization ability and contributes to online prediction. Shiming He [40] proposed a novel troubleshooting method called the DEPSO-RVM. In the proposed method, a particle swarm optimization (PSO) algorism and differential evolution (DE) algorism were employed to improve the RVM and four diagnostic models were developed to verify its effectiveness. The experimental results showed that the DEPSO-RVM had higher accuracy, which is conducive to life predictions.

Weilun Geng [41] explored the best prediction for fuel cell prediction models based on relevance vector machines with different kernel functions. Four different RVM prediction models were established for comparative analysis and the forecasting effect was discussed. It was found that the mixed kernel function had the highest accuracy and excellent learning ability. Kui Chen et al. [42] used vehicle operation data to predict the PEMFC based on multi-kernel relevance vector regression (MRVR) and the whale optimization algorithm (WOA). This method proved to be more accurate than the single kernel function method by comparing other algorithms. The prediction block diagram of PEMFC degradation using the WOA-MRVR is given in Figure 4.

### 2.6. Gaussian Process Regression

GPR is a nonparametric model that uses the Gaussian process (GP) prior to data regression analysis. The assumption of the GPR model consists of prior noise and Gaussian processes and its solution is based on Bayesian inference.

Because it is difficult to extract learning data from PEMFC high-frequency noise for training and learning using the current method, Yucen Xie [43] proposed a remaining useful life (RUL) prognostication technique combining singular spectrum analysis (SSA) and a deep Gaussian process (DGP). The first step of the SSA-DGP model was to apply SSA to remove noise data and use DGP to learn the degradation characteristics for life prediction after obtaining the processed data. The experimental results proved that this approach had better accuracy, could avoid excessive fitting, and had a respectable effect. The DGP prediction model training is shown in Figure 5 [43].

Huiwen Deng [44] designed a GPR modeling framework based on a variational auto-encoded deep Gaussian process (VAE-DGP) and a sparse pseudo-input Gaussian process (SPGP) to forecast the degradation trend of PEMFCs and deal with model uncertianties. Static and dynamic aging tests were conducted with electric tension and delivered the power of the stack as health determinants. The experiments showed that this method could be applied to less data and had higher precision, making it superior to the other data models.

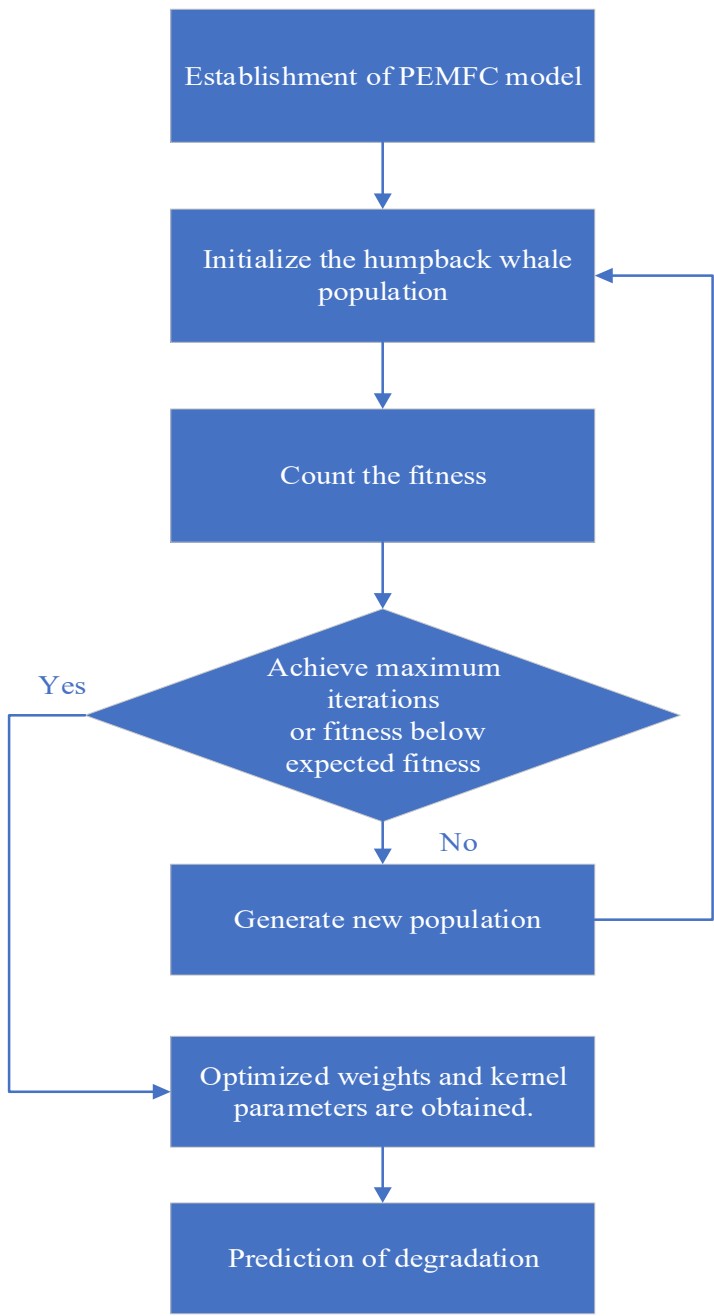

**Figure 4.** The prediction block diagram of PEMFC degradation using WOA-MRVR.

### 2.7. Extreme Learning Machine

Compared with the traditional neural network, the ELM's advantage is that the weight calculation is from the hidden layer to the output layer without iteration so it has a faster learning ability and training speed [45]. Yi-Peng Xu et al. [46] designed an improved ELM model for identifying the PEMFC stack system parameters. In addition, the ELM algorithm had rapidity and significant generalization performance [47].

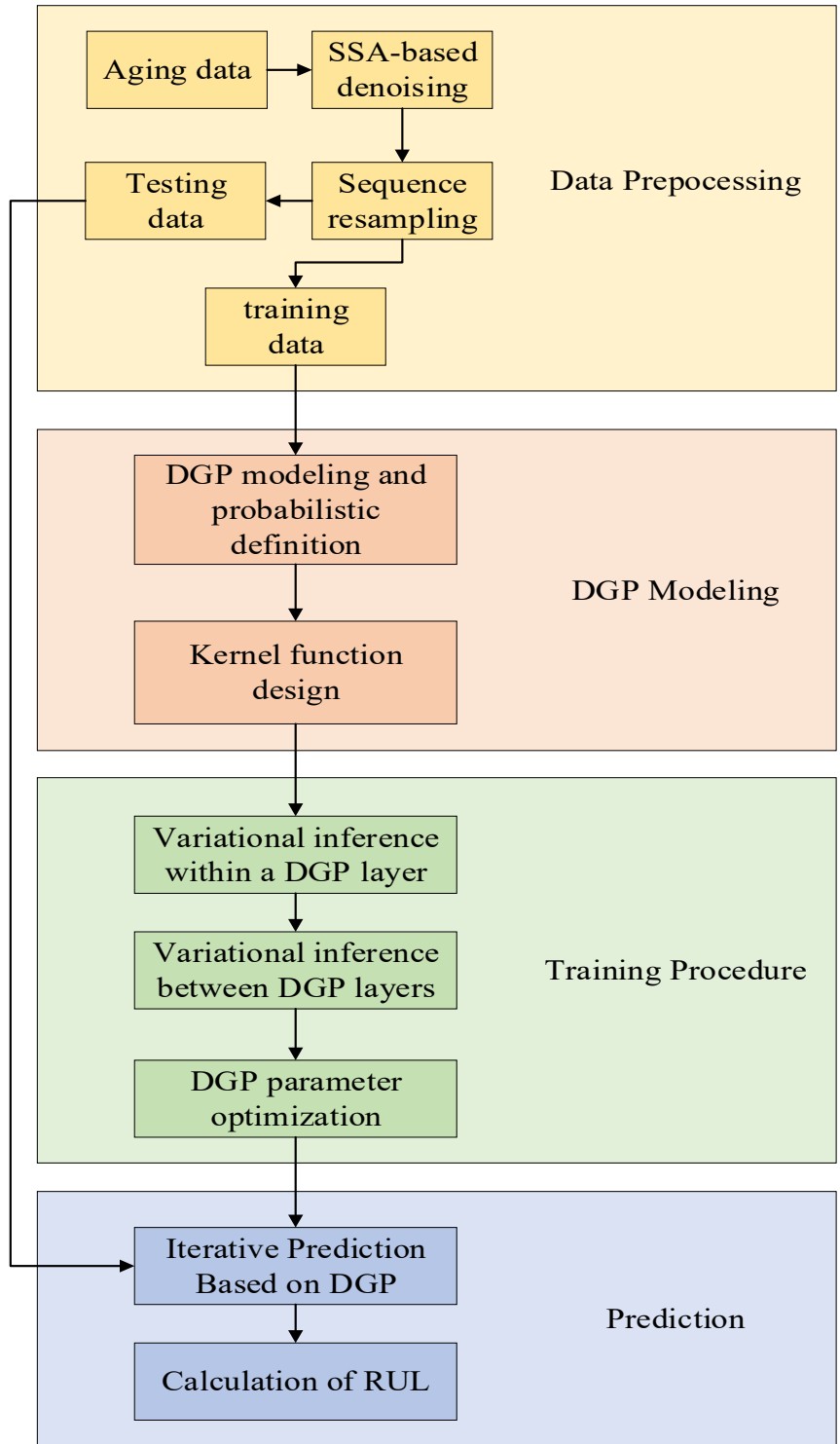

**Figure 5.** PEMFC degradation prediction framework based on SSA-DGP.

Xie Y [48] combined the deep belief network (DBN) and ELM to accurately predict PEMFC performance degradation. This method could not only accurately extract the nonlinear characteristics from the data but also improved the prediction reliability. The developed degradation prediction method had accurate and stable prediction performance in different sample sizes and prediction ranges. The deficiency was that the method was not applied to PEMFC life prediction under dynamic conditions. The overall prediction framework is shown in Figure 6.

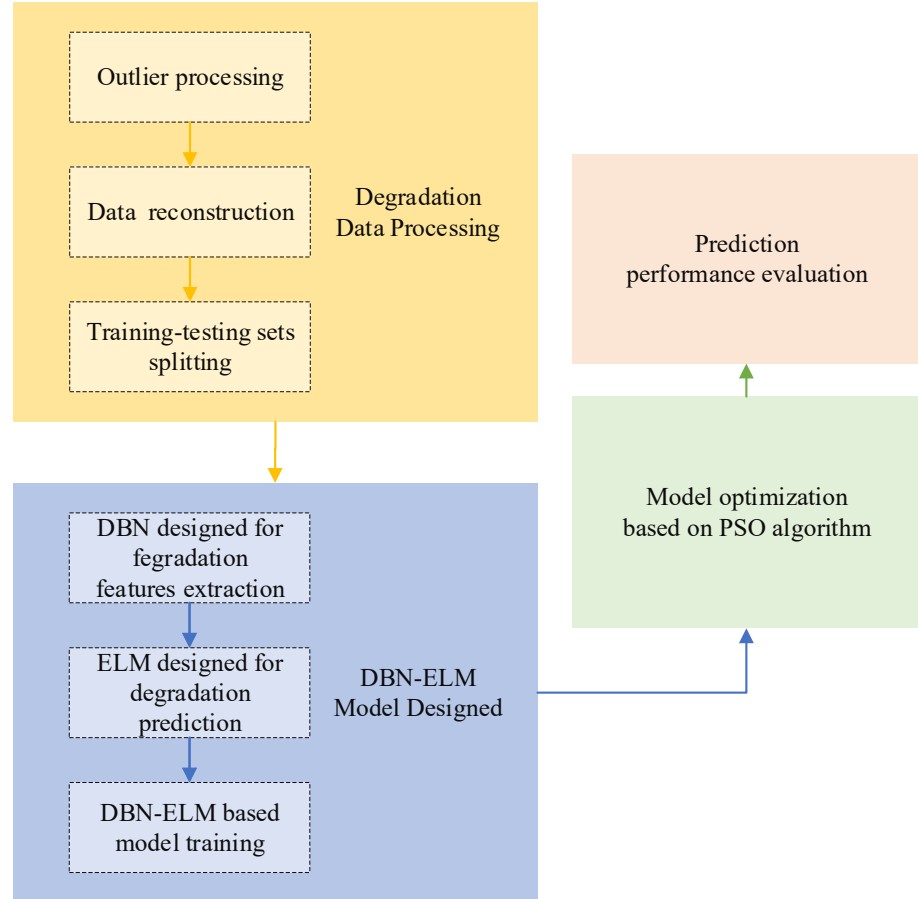

**Figure 6.** DBN-ELM overall forecasting framework.

Boyan et al. [49] developed a hybrid strategy with extreme learning machine and meta-heuristic algorithms to extract multiple unknown parameters of the solid oxide fuel cell model. Through simulation analysis, this method was more stable and had higher prediction accuracy than the traditional ELM. Xuexia Zhang and her colleagues [50] integrated discrete wavelet transform and adaptive differential evolution algorithms into an extreme learning machine to predict the PEMFC's life. The running data under constant currents were compared with the ELM. The experimental results showed that the method had better robustness and faster convergence speed and was more suitable for small sample predictions, but that variable load predictions were still a challenge.

*2.8. Digital Twin*

Safa Meraghni [51] constructed a digital twin model to predict the RUL of the PEMFC and updated the online data by connecting the digital twin side to the physical side and the number of data had little effect on the accuracy of the results. The digital twin prediction method had a breakthrough in online predictions and combined with the health management system could have good application prospects.

**3. Model-Based Approaches**

The model-driven method relies on the internal reaction mechanism of the batteries to establish a comprehensive electrochemical formula to obtain accurate life predictions. This method requires fewer process data [52] but its accuracy depends on the consistency between the established model and the actual system [53–55]. Model-driven methods include particle filter (PF), Kalman filter (KF), degradation mechanism, and empirical.

### 3.1. Particle Filter

As a nonlinear filtering algorithm based on Bayesian estimation, PF can eliminate various error effects including systematic errors and random errors when processing data [56] and can include unknown states through its degradation into physical models. This method can predict the remaining lifetime of the system by continuously drawing the probability distribution of possible degradation states.

Considering the effects of load current and other factors, Kui Chen [57] used the grey neural network to establish the recession model, and a particle swarm optimization algorithm was employed to optimize the weights and thresholds; then, iterative training was carried out based on the different moving window sizes. Through the recession experiment of the battery under three working conditions of static current, dynamic load current, and postal fuel cell electric vehicle, they concluded that the proposed method could obtain a satisfactory prediction accuracy in a small amount of data.

Yujie Cheng [58] combined regularized particle filter (RPF) with least squares support vector machine (LSSVM), which could accurately capture the nonlinearity in the data set and carry out a better assessment of the degradation trend. The predictability of the RPF and PF methods was compared to verify the effectiveness of this method. This method could provide the uncertainty characteristics of the RUL while also predicting it.

Mayank Shekhar Jh et al. [59] proposed a solution for the prediction of industrial PEMFCs. Combining PF's advantages with the fault indicator derived from the Bond Graph model, it was used to predict electrochemical parts and proved to be more effective in degradation tests than the extended Kalman filter.

### 3.2. Kalman Filtering

KF is a kind of algorithm for the optimal estimator of the system state through import and export to observe the data of the linear system. Due to the observational data containing the influences of noise and disturbances in the system, the sharp estimation could also be looked upon as a filtered process [60]. However, the accurate estimation of battery life using the Kalman filter depends largely on accurate battery modeling and its online model parameter estimations [61].

Based on KF, Yunjin Ao [62] proposed a frequency-domain Kalman filter(FDKF)to forecast the remaining service life. The effectiveness of the method was verified using an example, and it was superior to the other methods based on KF. Mathieu Bresse et al. [63] employed an extended Kalman filter (EKF) to estimate the health and degradation trends. The method was applied to the experimental data of long-term battery heap tests to verify their validity and the error effect was small. Zhuo Wang [64] demonstrated the use of cell-level state estimation technology in large-scale battery energy storage systems using experimental data from a 2 MW, 1 MWh battery energy storage system. It proved that the state of charge (SOC) estimation of the double sigma point Kalman filter (DSPKF) could obtain accurate estimation results with a smaller computational burden. Kui Chen et al. [65] combined the unscented Kalman filter (UKF) algorithm with a voltage degradation model for fuel cell degradation trend predictions. The proposed algorithm was also guaranteed in practical applications for the degradation trend of the battery under different working conditions.

### 3.3. Degradation Mechanism Model

The degradation mechanism model obtains life prediction parameters of the PEMFC through the aging internal mechanism, which has strong robustness and superiority but cannot forecast the PEMFC life under dynamic loads.

Kui Chen [66] considered the effect of the PEMFC loading current and proposed a hybrid degradation model based on wavelet analysis, ELM, and a genetic algorithm (GA). Through the verification of the actual data of the electric vehicles, it was found that the model was faster than the traditional prediction method and met the calculation conditions of online measurement in practice and had higher accuracy. The diagram of the optimized

GA-ELM using wavelet analysis is shown in Figure 7. Manik Mayur [67] proposed a physical-based regression model that used Pt degradation as a degradation mechanism to predict the durability of the PEMFC. By comparing and analyzing the degradation of two different driving cycle batteries and observing the dissolution rate of platinum, this rate could be used as the monitoring index of the PEMFC to obtain a healthy PEMFC state.

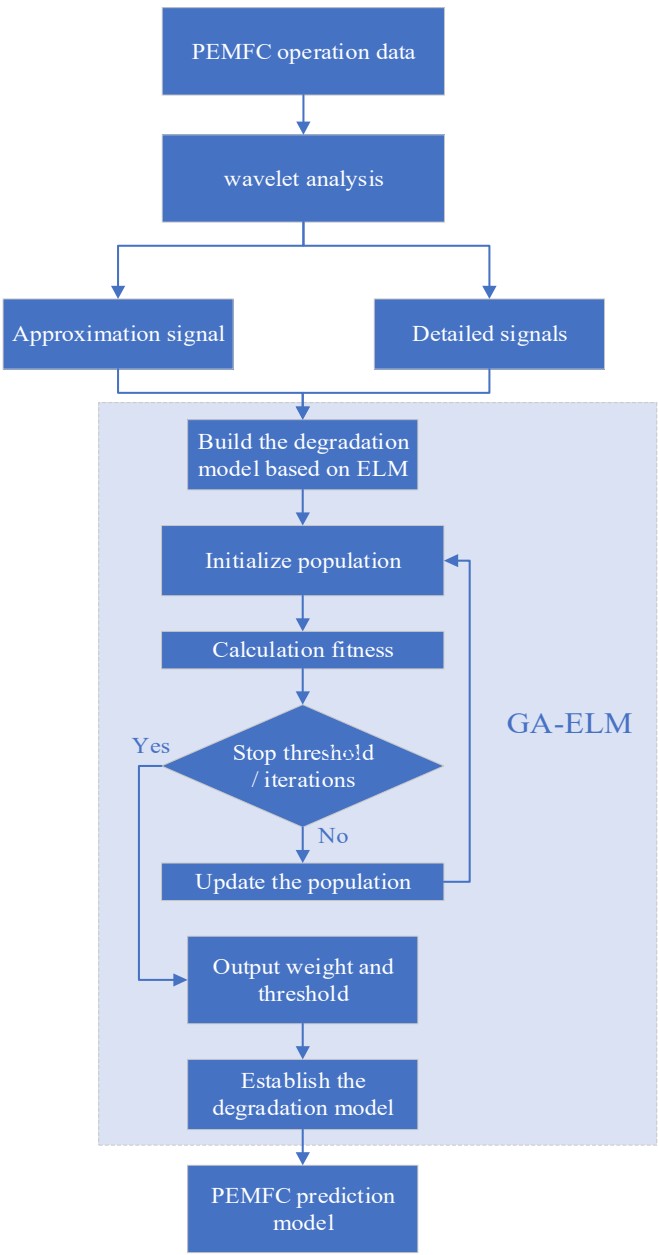

**Figure 7.** GA-ELM flow chart optimized by wavelet analysis.

*3.4. Empirical Model*

The mapping relations between the aging parameters and experimental conditions are realized by introducing parameter identification into an empirical model with high accuracy and reliability. Marvin Messing et al. [68] carried out load cycle and start/stop cycle experiments and the consequences were used to build the empirical durability model of cathode electrodes during the operation of fuel cell vehicles, which could be further integrated with the membrane durability model so as to comprehensively predict the life and performance recession of fuel cells under wide application conditions and system

designs. Alexander Kneer [69] proposed a semi-empirical electrochemically active surface area loss model to predict the dissolution of Pt and estimate the degradation of catalyst for fuel cells. The experimental results showed that the semi-empirical loss model could measure the corresponding loss in different voltage cycles and had high accuracy for degradation predictions. Mingyang Ou [70] proposed a semi-empirical model method for accurate PEMFC life predictions. The model considered various polarization behaviors and introduced parameter identification to process complex data in the model. The feasibility of the model was verified by the PEMFC data sets.

## 4. Hybrid Approaches

The structure of data-driven approaches is simple, but it depends on the quality and quantity of data. The model-driven method has high forecast accuracy but the structure is complex and it is difficult to fully consider the reaction mechanism within the PEMFC. Therefore, how to combine the advantages of both for accurate life predictions is of great significance. Combining model and data-driven methods with different hybrid strategies can achieve more accurate life predictions, but the structure is more complex and the amounts of the calculations are larger. The hybrid approaches overcome the boundedness and shortcomings of the model and data-driven methods for measuring residual lifetime [71]. Previously, hybrid approaches were applied in other areas [72–75].

Model-driven adaptive KF was fused with data-driven NARX to predict the PEMFC life by Rui Pan et al. [76]. Two groups of aging data under different conditions were used for the test. The comparative analysis showed that this method could capture the overall degradation trend information and detailed degradation information and had higher accuracy than other prediction methods. Liu Hao [77] introduced a novel comprehensive analysis method to analyze the aging tendency and residual PEMFC life under various current loads. On this basis, a machine learning algorithm founded on an evolutionary algorithm and adaptive fuzzy logic was proposed. Moreover, semi-empirical degradation and UKF were used to evaluate the residual lifetime of the system. The test results showed that compared with the previous method based on the integrated model, the combined prediction method had higher prediction accuracy and faster prediction speed for battery residual life. Chu Wang et al. [78] employed a new fusion prediction technology to analyze the health index of FC using an aging model and moving window method, then utilized symbol-based LSTM to predict the degradation trend in the health indicators. The experimental results showed that the proposed method could reflect the voltage change in FC storage over time, with a large range and high accuracy. Huicui Chen et al. [79] introduced a new idea for machine learning predictions based on the operating parameters. By predicting voltage consistency under different combinations of operating parameters, it could accurately predict the online PEMFC life. In order to solve the degradation prediction problem of the PEMFC in the frequency domain, Yunjin Ao et al. [80] proposed a voltage degradation model and FDKF joint drive method to predict PEMFC degradation. Degradation experiments were conducted at a constant current and dynamic current to verify the prediction performance under different working conditions. Compared with other prediction methods, the proposed method had better accuracy and robustness. Penghao Wang [81] introduced polarization resistance and combined the degradation model with PF to predict the future PEMFC degradation trend (FDT) and the RUL. Compared with the commonly used aging model, the errors were smaller and the accuracy was higher. Daming Zhou et al. [82] proposed a new stability prediction method to forecast PEMFC life. Firstly, the physical aging model (PAM) was used to analyze the degradation process of the original PEMFC to eliminate its unstable changes. Then, they used the autocorrelation function (ACF), partial ACF, Akaike, and other information criteria to determine the order of the autoregressive moving average model, and then the autoregressive and moving average (ARMA) method was used to filter the linear components in smooth times series. Finally, the residual nonlinear damping models were utilized to train the delayed neural network to obtain the ultimate forecast effect. The prediction results showed that the innovative

PAM-ARMA-TDNN (time delay neural network) method had higher forecast accuracy and robustness and was more reliable in practical prediction work.

There are multi-data-driven and multi-model-driven approaches in the hybrid method. The multi-data driven approach is to weight or integrate various data-driven methods to improve the accuracy and robustness of remaining life predictions. Without establishing a complex battery model, the relationship between a battery's state and its external parameters can be automatically explored [83]. To accurately predict PEMFC life Chen et al. [84] proposed an aging prediction model based on a BP neural network and evolutionary algorithm. The degradation prediction model of the PEMFC was established using the BP neural network. The degradation model was optimized using the mind evolutionary algorithm (MEA), PSO, and GA with a higher precision ratio. Rui Ma et al. [85] proposed a PEMFC performance data integration prediction method based on an LSTM recurrent neural network and autoregressive integrated moving average (ARIMA). The proposed LSTM-ARIMA method could accurately forecast the decomposition of the PEMFC via experiments, making online diagnostic control possible, as shown in Figure 8. Rui Ma [86] established the physical aging model of the PEMFC to reflect the internal aging parameters to predict PEMFC life. The nonlinear characteristics of the PEMFC were filtered using EKF, and the parameter updating problem was solved by the LSTM. The validity of the method was verified using static and quasi-dynamic test data. The experiment showed that the technique could accurately forecast the degeneration trend of the PEMFC output tension and aging parameters under different training stages. Focusing on the difficulties in online predictions, Penghao Wang [87] and other scholars proposed a nonlinear empirical degradation model and then employed PF to estimate the degradation state variables online to achieve the online prediction standard of the PEMFC. Taking the rated voltage as the new aging index, it could achieve superb life predictions in variable load and online predictions.

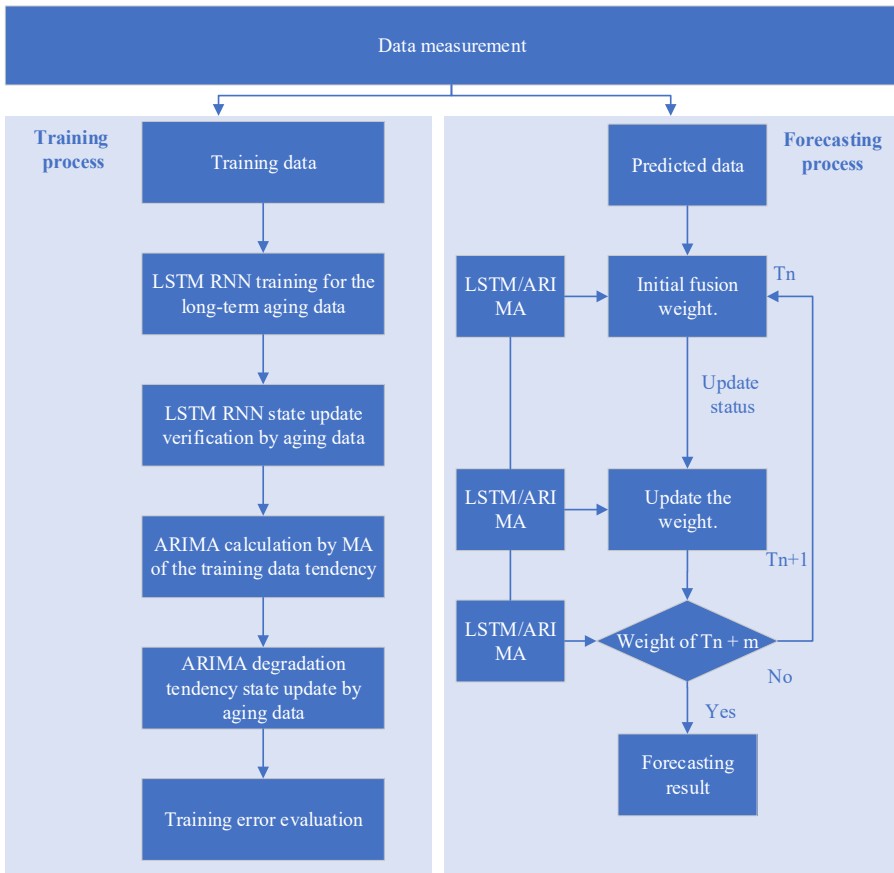

**Figure 8.** LSTM-ARIMA data-fusion prognostic model.

## 5. Prospects and Challenges of PEMFC

In this paper, the classification and research of lifetime prediction methods for the PEMFC are compared, analyzed, and summarized. The methodology is divided into data-driven, model-driven, and hybrid approaches. Data-driven approaches are simple and convenient without considering the internal degradation mechanism of the PEMFC but require large amounts of comprehensive actual operation data. Compared with data-driven approaches, model-driven approaches require less data, higher accuracy, and wider application. However, it is difficult to predict the life of the research object without knowing its internal structure and reaction mechanisms, and model-driven approaches are more complex in their modeling. Hybrid prediction approaches have higher accuracy than the aforementioned two methods but make the calculations more complicated. These approaches have certain reliability but are far from being applied in practice. In the area of PEMFC life predictions, online forecasts are still the key issue. However, considering the calculation time, robustness, and accuracy required for online predictions, the current methods cannot be directly applied to practice. The operating conditions of PEMFCs in electric vehicles are complex, and the life prediction methods cannot be verified according to the constant load conditions in general experiments. It is necessary to demonstrate the reliability of this method under complex dynamic conditions. According to the current researches, the primary aging indicators are voltage and power, which cannot accurately predict PEMFC life under dynamic operating conditions in electric vehicles. Consequently, it is particularly important to extract aging indicators that can be utilized for accurate online PEMFC life predictions with a low calculation burden under dynamic conditions. With the wide application of the Internet of Vehicles in the automotive field, future life predictions of the PEMFC are closely related to algorithms, and the parameters, structures, and degradation indexes in life predictions tend to be automatically selected through those algorithms. In prediction methods, especially for the data-driven and empirical models, the use of an integrated framework could greatly improve the accuracy and reliability of the prediction results. In addition, the advantage of data-driven approaches is that they train and analyze trends, whereas the advantage of model-driven approaches is that they extract aging indicators. The hybrid method formed by the combination of the two methods has great advantages in dynamic conditions, which is helpful for the accurate application of life predictions in practice under the premise of solving the huge calculation burden.

## 6. Conclusions

In this paper, the life prediction methods of the PEMFC are reviewed and classified. The data-driven method and the model-driven method each have their advantages and disadvantages, which are the basis of PEMFC life prediction methods and have been gradually adopted to solve practical problems. In this paper, the three methods for life prediction are compared and discussed, the related research is discussed in detail, and a more characteristic outline is drawn. The challenges faced by the current research and future research directions are summarized, which could have a guiding role in subsequent life prediction method research. However, this paper only classifies the life prediction methods for the PEMFC and is not suitable for application in other batteries. The development process of the life prediction methods has not been summarized as the focus of this work is on recent research results. In the follow-up studies, the life prediction methods of the PEMFC will be more comprehensively summarized and more valuable research directions will be explored.

**Funding:** This research was funded by the Innovation Team at the Institution of Higher Education in Chongqing, grant number: CXQT21027; Chongqing Talent Scheme, grant number: cstc2021ycjh-bgzxm0261; the Natural Science Foundation of Chongqing, China, grant number: cstc2021jcyj-msxmX0464; the Scientific Research Foundation of the Chongqing University of Technology, grant number: 2021ZDZ004; the National Natural Science Foundation of China, grant number: 52177210; and the Natural Science Program of Shandong Province, grant number: ZR2020ME209.

**Institutional Review Board Statement:** Not applicable.

**Informed Consent Statement:** Not applicable.

**Data Availability Statement:** Not applicable.

**Acknowledgments:** The authors are grateful to the Chongqing Municipal Education Commission, the Chongqing Science and Technology Bureau, the National Natural Science Foundation of China, the Department of Shandong Science and Technology, and the Chongqing University of Technology.

**Conflicts of Interest:** The authors declare no conflict of interest.

## Abbreviations

| | | | |
|---|---|---|---|
| **ANFIS** | Adaptive neuro-fuzzy inference system | **ARMA** | Autoregressive and moving average |
| **ARIMA** | Autoregressive integrated moving average | **ACF** | Autocorrelation function |
| **BP** | Backpropagation | **DSPK** | Double sigma point Kalman filter |
| **DBN** | Deep belief network | **DE** | Differential evolution |
| **DGP** | Deep Gaussian process | **EKF** | Extended Kalman filter |
| **ESN** | Echo state network | **ELM** | Extreme learning machine |
| **FDKF** | Frequency-domain Kalman filter | **FDT** | Future degradation trend |
| **GP** | Gaussian process | **GA** | Genetic algorithm |
| **GPR** | Gaussian process regression | **KF** | Kalman filter |
| **LSSVM** | Least squares support vector machine | **LSTM** | Long- and short-term memory network |
| **MRVR** | Multi-kernel relevance vector regression | **MIMO** | Multiple-input multiple-output |
| **MEA** | Mind evolutionary algorithm | **NOE** | Nonlinear output error |
| **NARX** | Nonlinear autoregressive exogenous | **NSD** | Navigation sequence-driven |
| **PSO** | Particle swarm optimization | **PAM** | Physical aging model |
| **PF** | Particle filter | **RUL** | Remaining useful life |
| **PEMFC** | Proton exchange membrane fuel cell | **RPF** | Regularized particle filter |
| **RVM** | Relevance vector machine | **SPGP** | Sparse pseudo-input Gaussian process |
| **SSA** | Singular spectrum analysis | **SOC** | State of charge |
| **UKF** | Unscented Kalman filter | **VAE** | Variational auto-encoded |
| **WOA** | Whale optimization algorithm | | |

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
