# Peer review of "A Review of Life Prediction Methods for PEMFCs in Electric Vehicles"

_sustainability, doi:10.3390/su14169842_

Round 1
Reviewer 1 Report
The content of this article is good. It is interesting to read the article. However, the following suggestions should be incorporated before the publication.
1. The abstract of this article should be modified. The importance of this review article is not highlighted in the current version.
2. The introduction is very simple and does not elaborate on the technical findings in the previous literature. The reviewer understands that the article type is a review article, however, the presented content in the introduction is too general.
3. Many abbreviations are there. Please create a nomenclature section and list out all the abbreviations.
4. The data-driven approaches and model-based approaches are discussed in detail in this article. Few research articles have already discussed these methods in their article as a part of their review. Please indicate the main authors' contribution to this article.
5. Section 5 could be divided into two parts such as "technical challenges" and "conclusion". Please create a separate section and modify the content.
6. Few articles discussed about the reliability analysis to predict the lifetime of the fuel cell. I believe that the authors didn't consider this topic. Please let me know why it is not included in this review article.
Minor correction:
1. A very limited article from 2022. Kindly include more recent articles.
2. Quality of the figures should be improved. Are those classifications (Fig. 1,2,4,5 etc) originally developed by authors or taken from existing literature survey? If it is taken from the existing literature, please cite the reference properly.
Author Response
Reviewer#1, Concern # 1:
The abstract of this article should be modified. The importance of this review article is not highlighted in the current version.
Author response:
Thank you very much for your suggestion. After discussion, we revised the abstract and highlighted the importance of this paper.
Reviewer#1, Concern # 2:
The introduction is very simple and does not elaborate on the technical findings in the previous literature. The reviewer understands that the article type is a review article, however, the presented content in the introduction is too general.
Author response:
Thank you very much for your suggestion. We revised the introduction and elaborated on the technical findings in the previous literature.
Reviewer#1, Concern # 3:
Many abbreviations are there. Please create a nomenclature section and list out all the abbreviations.
Author response:
Thank you for your vital advice. We added the abbreviation form after the introduction to make it easier for readers to read.
Reviewer#1, Concern # 4:
The data-driven approaches and model-based approaches are discussed in detail in this article. Few research articles have already discussed these methods in their article as a part of their review. Please indicate the main authors' contribution to this article.
Author response:
We appreciate your careful reading and extremely detailed guidance.We have revised the content and highlighted the importance of data and model drivers.
Reviewer#1, Concern # 5:
Section 5 could be divided into two parts such as "technical challenges" and "conclusion". Please create a separate section and modify the content.
Author response:
We appreciate your careful reading and extremely detailed guidance. We divide the last part into two parts to make the structure of the article clearer.
Reviewer#1, Concern # 6:
Few articles discussed about the reliability analysis to predict the lifetime of the fuel cell. I believe that the authors didn't consider this topic. Please let me know why it is not included in this review article.
Author response:
Thank you very much for your careful review of this article. Your suggestion is very helpful to us. We are sorry that this reliability analysis method is not taken into account. As you said, there is very little literature that uses reliability analysis for prediction. This method is really very important, and we will consider it systematically and address it in the next work. Thank you very much for the loopholes you pointed out.
Reviewer#1, Concern # 7:
A very limited article from 2022. Kindly include more recent articles.
Author response:
We sincerely appreciate your comments. The questions you mentioned are very important to our manuscript. We have changed the literature and introduced more latest research into this paper. Your valuable suggestions are of great help to us.
Reviewer#1, Concern # 8:
Quality of the figures should be improved. Are those classifications (Fig. 1,2,4,5 etc) originally developed by authors or taken from existing literature survey? If it is taken from the existing literature, please cite the reference properly.
Author response:
Thank the reviewers for their comments. We annotate the source of the referenced graphs and fine-tune some graphs.

Reviewer 2 Report
The author did a very good job in providing the review of all methods available in literature.
1. I feel the authors could explain more in introduction regarding why PEMFC and how PEMFC degrades what is PEMFC degradation index etc.,.
2. In conclusions the author mentioned "The combination of the two methods will be a main direction for future life prediction methods." I am confused here, because the author mentioned that the hybrid (as per my understanding combination of two methods) still has issues. But the author at the end mentions hybrid as the solution. This is misleading. Can you please elaborate on the Prospect and Summary section.
Author Response
1.The authors could explain more in introduction regarding why PEMFC and how PEMFC degrades what is PEMFC degradation index etc...
Author response:
We sincerely thank you for your criticism and guidance after careful reading of our manuscript. We agree with you very much. Therefore, we further introduced the failure mechanism and degradation index, and revised the literature.
2. In conclusions the author mentioned "The combination of the two methods will be a main direction for future life prediction methods." I am confused here, because the author mentioned that the hybrid (as per my understanding combination of two methods) still has issues. But the author at the end mentions hybrid as the solution. This is misleading. Can you please elaborate on the Prospect and Summary section.
Author response:
Thank you very much for your valuable suggestions. It is mentioned in the summary that the hybrid method is really difficult to use in the actual tram because of its complex structure and large amount of calculation. However, compared with the data-driven and model-driven methods, the hybrid method has higher accuracy and integration advantages of the first two methods, so that it can be more suitable for actual complex conditions. We reorganized the statements in the outlook and apologize for the misleading things about this error.

Reviewer 3 Report
This paper reviews the proton exchange membrane fuel cell life prediction methods for electric vehicles. The subject of the manuscript falls within the scope of the journal. This paper is a good contribution to this area of research. The following issues should be addressed to improve the quality of this paper:
a. The contribution of this research needs to be better pointed out.
b. It is recommended that a keywords co-occurrence map to be included in the introduction section.
c. Once a term has been abbreviated, it is recommended the use of the abbreviated form consistently from then on.
d. The authors should point out the limitations of the study.
e. Some acronyms are used without being defined before (line 125, 152).
f. The last section should better highlight the contribution of this investigation.
Author Response
1.The contribution of this research needs to be better pointed out.
Author response:
Thank you very much for your correction. At the end of the introduction, we revised it to express the contribution of this study as clearly as possible.
2.It is recommended that a keywords co-occurrence map to be included in the introduction section.
Author response:
We appreciate your careful reading and extremely detailed guidance. We added an appropriate introduction to the keywords in the introduction to make the content feel fuller.
3.Once a term has been abbreviated, it is recommended the use of the abbreviated form consistently from then on.
Author response:
Thank you very much for your detailed review. We made a rigorous revision of the abbreviations of the full text to make sure there was a complete noun in front.
4.The authors should point out the limitations of the study.
Author response:
We sincerely appreciate your comments. The questions you mentioned are very important to our manuscript. As a missing part of our manuscript, we think adding relevant content is very helpful to improve the quality of our manuscript. At the end of this paper, we add the shortcomings of this paper, and believe that this paper takes PEMFC as the research object, and does not confirm that the prediction method is applicable to other types of batteries. The paper is based on the literature issued in recent years, and does not summarize the development history of life prediction method.
5.Some acronyms are used without being defined before (line 125, 152).
Author response:
The noun abbreviations appearing in lines 125 and 152 are already mentioned in the front and have complete nouns. Thank you very much for your careful review, we will be more rigorous to address these basic errors.
6.The last section should better highlight the contribution of this investigation.
Author response:
Thank you very much for your valuable feedback. We revised the end of the manuscript and appropriately revised the contribution of this paper.

Round 2
Reviewer 1 Report
Thank you very much for your revision. The revised article looks good.